# Specific Alcohol-Responsive Photonic Crystal Sensors Based on Host-Guest Recognition

**DOI:** 10.3390/gels9020083

**Published:** 2023-01-18

**Authors:** Xiaolu Cai, Xiaojing Zhang, Jing Fan, Wenxiang Zheng, Tianyi Zhang, Lili Qiu, Zihui Meng

**Affiliations:** 1School of Chemistry and Chemical Engineering, Beijing Institute of Technology, Beijing 100081, China; 2Quality Control Center Department, Sinosteel Luoyang Institute of Refractories Research Co., Ltd., Luoyang 471039, China

**Keywords:** photonic crystals, β-cyclodextrin, alcohol sensors, volatile organic compounds detection

## Abstract

A photonic crystal material based on β-cyclodextrin (β-CD) with adsorption capacity is reported. The materials ((A-β-CD)-AM PC) consist of 3D poly (methyl methacrylate) (PMMA) colloidal microsphere arrays and hydrogels supplemented with β-cyclodextrin modified by acryloyl chloride. The prepared materials are then utilized for VOCs gas sensing. The 3D O-(A-β-CD)-AM PC was used to detect toluene, xylene, and acetone and the response was seen as the red-shift of the reflection peak. The 3D I-(A-β-CD)-AM PC was used to detect toluene, xylene, and acetone which occurred redshifted, while methanol, ethanol, and propanol and the peaks’ red-shifting was observed. However, among these, methanol gave the largest red-shift response The sensor has broad prospects in the detection of alcohol and the detection of alcohol-loaded drug releases in the future.

## 1. Introduction

Harmful gases and volatile organic compounds (VOCs) reduce air quality and damage human health, which increases the demand for the research of high-performance gas sensors [1,2,3,4]. VOCs are ubiquitous in modern building materials, furniture, house decoration, and spoiled food. Prolonged exposure to VOCs can irritate the skin, eyes, and respiratory tract, and cause cancer and leukemia [5,6,7,8]. In modern society, most people spend 70–90% of their time indoors. According to the World Health Organization, millions of people die from indoor air pollution every year. Therefore, real-time detection of air quality is something to be taken seriously. Currently, many types of gas sensors have been developed such as chemiresistive gas sensors [9,10], gasistors [11], solid electrolyte-type gas sensors [2,12], surface acoustic wave sensors [13], mid-infrared sensors [14], optical fiber sensors [15], and photonic crystal sensors [16,17]. Among them, photonic crystals (PCs) obtain incomparable advantages as artificial materials. PCs are periodic structures composed of materials with different dielectric constants [18,19]. Compared to other sensors, PC sensors have attracted widespread attention because of immediate stimulus responses [16,17,20,21,22,23,24,25], naked-eye detection, and without power. Their unique structural colors change correspondingly with the lattice spacing and effective refractive index (ERI) of the PCs when physical or chemical stimuli occur.

To magnify the effect of stimulus-response, PCs based on 3D Poly (methyl methacrylate) (PMMA) colloidal microspheres require auxiliary materials such as hydrogels and elastomers [26,27,28]. Hydrogels are three-dimensional networks composed of cross-linked hydrophilic macromolecules. The functions of hydrogels can be designed according to the type of detected gas. Hydrogel photonic crystals respond very well to VOCs. One is because VOCs easily enter the hydrogel interior to change its ERI, and the other is because VOCs change the volume and shape of the hydrogel, causing a large change in the lattice spacing of the hydrogel photonic crystal and producing the change of bright, naked-eye visible structural color.

β-cyclodextrin (β-CD) is a semi-natural product of starch fermentation, which has good biocompatibility and chemical stability. As shown in Figure 1a, it is a cyclic oligosaccharide composed of seven D-glucopyranoside units through the action of α-1,4 glycosidic bonds [29]. It can form host–guest complexes through cavities with guests, such as inorganic ions or organic molecules with simple structures and small sizes, because of external hydrophilicity and internal hydrophobicity [30]. This peculiar structure can be used in drug delivery [31], pharmaceutical contaminants sensors [32], adsorptions of organic compounds [33] and herbicides [34], assisted extraction of compounds from plants [35,36], and capture of gas [37,38]. A single Debye-type relaxational absorption by ultrasonic relaxation method was found only when both β-CD and the alcohols coexisted, which proved there was a chemical dynamic equilibrium [39]. The same phenomena were also found in the aqueous solution of amides, esters, benzene, and aromatic compounds [40,41,42]. β-CD has a strong adsorption capacity. Chen used β-CD modified silver nanoparticles for naked-eye detection of aromatic isomers, and the present detection limit for different isomers of aromatic compounds is 5 × 10^−5^ M [43].

Herein, a rapid photonic crystal sensor based on β-CD was developed. The sensor ((A-β-CD)-AM PC) is composed of β-CD modified by acryloyl chloride and 3D poly (methyl methacrylate) colloidal microspheres (Figure 1). (A-β-CD)-AM PC is used for naked eye detection of VOCs due to their bright structural color. 3D O-(A-β-CD)-AM PC was used to detect toluene, xylene, and acetone, and 3D I-(A-β-CD)-AM PC was used to detect toluene, xylene, acetone, methanol, ethanol, and propyl alcohol. The 3D I-(A-β-CD)-AM PC had the best detection performance for alcohols, especially on methanol. (A-β-CD)-AM PC has the prospect of application in the detection of VOCs and drugs dissolved in alcohol.

## 2. Results and Discussion

### 2.1. Characterization of A-β-CD

The -OH of two primary alcohol hydroxyl groups on the smaller diameter face of the β-CD cavity was fractured and β-CD was combined with acryloyl chloride in the presence of KOH as shown in Figure 1a. Synthetic cyclodextrins contain -C=C- bonds to act as cross-linking agents in hydrogel components by free radical polymerization [44] as shown in Figure 1b. The obtained A-β-CD was dissolved in dimethyl sulfoxide (DMSO) for NMR analysis. The hydrogen NMR spectrum is shown in Figure 2a. ^1^H NMR (400 MHz, DMSO-d6) δ:6.40–6.18 (m, 1H), 4.84 (s, 3H), 3.64 (s, 3H), 3.55 (s, 1H), 2.09 (s, 0H) ppm. The carbon nuclear magnetic resonance spectrum is shown in Figure 2b. ^13^C NMR (101 MHz, DMSO-d6), δ:167.39, 131.23, 129.95, 102.38, 81.98, 73.53, 72.88, 72.49, 63.25, 60.38 ppm. The FTIR of A-β-CD measured by potassium bromide tablet is shown in Figure 2c. FTIR (KBr), v/cm^−1^: 3457.5, 2921.6, 1725.9, 1624.3. Among them, 1725.9 cm^−1^ is the -C=O- stretching vibration absorption peak, and 1624.3 cm^−1^ is the -C=C- stretching vibration absorption peak.

The structure and functional groups of the synthesized A-β-CD characterized by NMR spectrum and FTIR were confirmed. Compared with the literature [45], it was confirmed that the synthesized product was indeed the target product.

### 2.2. Characterization of 3D O-(A-β-CD)-AM PC and 3D I-(A-β-CD)-AM PC

Poly(methyl methacrylate) (PMMA) colloidal microspheres with different particle sizes were prepared by adjusting the amount of MMA and KPS and had good sphericity, dispersion, and uniformity. The reflection peak (*λ_max_*) of the PCs changed from 400 nm to 625 nm as the PMMA particle size changed from 165 nm to 255 nm. PMMA colloidal microspheres were arranged into a 3D closely face-centered cubic (FCC) structure by a vertical deposition self-assembly method as shown in Appendix A.

The most suitable hydrogel formulation shown in Appendix A for combining with 3D PMMA arrays was 1.20 g A-β-CD, 1.60 g AM, 0.06 g BIS, 8 mL ultrapure water, and 5 μL DEAP. As shown in Figure 3a, PMMA arrays with different particle sizes combined well with hydrogels and exhibited bright structural colors. In Figure 3b, the structural colors of the 3D O-(A-β-CD)-AM PCs covered almost the entire visible light region. Optical characteristics of 3D O-(A-β-CD)-AM PCs were characterized by a spectrometer (Figure 3c–h). The hydrogel precursors penetrated the gap of the PMMA array by capillary force and replaced the air. The effective refractive index (ERI) of the hydrogel was greater than that of the air, so *λ_max_* changed from 440 nm to 737 nm as the *λ_max_* of the arrays changed from 400 nm to 625 nm.

The gap between larger colloidal microsphere arrays is larger, which helps to fill more hydrogels. About 225 nm and 240 nm PMMA arrays were selected for the VOCs sensor. As shown in Figure 4a, the gaps of the 225 nm PMMA array are filled with hydrogel and still have the FCC structure. 3D I-(A-β-CD)-AM PCs were obtained by etching 240 nm PMMA in toluene (Figure 4c). The diameter of the air hole was slightly smaller than the PMMA diameter. The pore structure after etching was uniform and had a large specific surface area. As shown in Figure 4d, the *λ_max_* of 3D O-(A-β-CD)-AM PCs was 648 nm, while the *λ_max_* of 3D I-(A-β-CD)-AM PCs was 582 nm. At the same time, the structural color changed from dark red to orange. The blue shift of the 3D I-(A-β-CD)-AM PC was caused by the replacement of PMMA with ERI of 1.48 by air with ERI of 1 and the reduction of the pore size of hydrogels. Optical properties of 3D PMMA arrays and 3D (A-β-CD)-AM PCs were determined using Bragg’s law, Equation (1):(1)$${\;\lambda }_{\max}=\sqrt{8/3}\left(d/m\right)\left(D/D_{0}\right){(\sum n_{i}^{2}v_{i}-\sin^{2}\theta )}^{1/2}\;$$
where *d* is the planar spacing; *m* is the order of Bragg diffraction, here *m* = 1; *D/D_0_* = 1 is the swelling ratio of the PCs; *θ* is the angle of the incident light and the normal of the crystal surface; *i* is the component that makes up the photonic crystal; *n* is the effective refractive index; *v* is the volume fraction of the component in the PC. According to the FCC structure, we assumed the *n_PMMA_* = 1.48, *n_(A-β-CD)-AM_* = 2.07, *v_PMMA_* = 0.74 and *v_(A-β-CD)-AM_* = 0.26 for 3D O-(A-β-CD)-AM PC and the *n_(A-β-CD)-AM_* = 2.07, *nair* = 1, *v_(A-β-CD)-AM_* = 0.26 and *v_air_* = 0.74 for 3D I-(A-β-CD)-AM PC. We obtained the refractive index of PMMA and air from the literatures [16,17]. The refractive index of (A-β-CD)-AM was measured by an Abbe refractometer. The experimental value is in good agreement with the theoretically calculated value and the relative errors were less than 1.6%.

### 2.3. Response of 3D O-(A-β-CD)-AM PCs to VOCs

As shown in Figure 5, 3D O-(A-β-CD)-AM PCs can be used to detect toluene, xylene, and acetone with the ERI of 1.4961, 1.4970, and 1.3588 respectively. When the gas content was different, the position and intensity of *λ_max_* changed correspondingly. As shown in Figure 5d–f, with the increase of xylene gas volume ratio, the position of *λ_max_* was red-shifted and the intensity of *λ_max_* decreased. In Figure 5a–c,g–h, the variation trend of the intensity and position of *λ_max_* is similar with the Figure 5d. Clearly, 3D O-(A-β-CD)-AM PC had the most obvious response to xylene gas with a red shift of 27 nm at the concentration of 820 ppm. 3D O-(A-β-CD)-AM PCs can detect xylene gas as low as 100 ppm. At the same time, it can detect toluene gas and acetone gas as low as 120 ppm and 80 ppm. As a comparison, when Wang from Meng’s group detected SO_2_ through opal photonic crystal cellulose membrane, the red shift of the reflection peak was only 7 nm [46]. Wang combined a 315 nm PMMA array with carboxymethyl cellulose to form a carboxymethyl cellulose photonic crystal film. The PMMA array was similarly bound to carboxymethyl cellulose as above but was thermally reinforced at 60 °C for 2 h. Therefore, β-CD improved the gas adsorption capacity of 3D O-(A-β-CD)-AM PC. In addition, the roughness on the surface of the PMMA colloidal microspheres due to the dissolution of the gas dissipated the energy of the reflected light resulting in a decrease in the intensity of *λ_max_*. The larger the gas volume ratio, the smaller the intensity of the *λ_max_*.

### 2.4. Response of 3D I-(A-β-CD)-AM to VOCs

3D I-(A-β-CD)-AM PC was prepared by etching PMMA colloidal microspheres with toluene and used to detect methanol, ethanol, and propanol. As shown in Figure 6a,b, as the concentration of methanol gas increased from 0 ppm to 1560 ppm, the position of λ_max_ was blue-shifted because the volume of the sensor shrunk and the lattice spacing got smaller, and the intensity of λ_max_ hardly changed. 3D I-(A-β-CD)-AM PC can detect methanol gas as low as 160 ppm. As shown in Figure 6c–f, the variation trend of the intensity and position of λ_max_ is similar with the Figure 6a,b. 3D I-(A-β-CD)-AM PC can detect ethanol gas and propanol gas as low as 100 ppm and 190 ppm. 

With the strong adsorption capacity of β-CD and alcohol-sensitive hydrogel framework, the sensor has excellent selectivity and reproducibility. Figure 7a is a schematic diagram of methanol entering the interior of the sensor. For convenience, only the hydrogel was drawn and the inverse opal structure after removing the 3D PMMA colloidal microsphere array was ignored. Methanol diffused from the air into the interior of the hydrogel. While other hydrophobic gases entered the cavity of β-CD more easily, they cannot pass through the barrier of an aqueous environment. Most interestingly, even acetone gas could enter the cavity of β-CD, but could not cause any volume shrinkage of the hydrogel. β-CD was like an amplifier, adsorbing more VOCs and triggering the hydrogel’s own response to gases. Selectivity is one of the important properties of a sensor. Toluene, xylene, acetone, methanol, ethanol, and propanol were detected to investigate the selective response of the 3D I-(A-β-CD)-AM PC to methanol vapor. As shown in Figure 7b, the *λ_max_* was blue-shifted in methanol, ethanol, and propanol, and *λ_max_* was red-shifted in toluene, xylene, and acetone. Alcohols easily form hydrogen bonds with the abundant -OH on β-CD in the hydrogel, so the reflection peak maximum shifts of methanol, ethanol, and propanol are much larger than those of toluene, xylene, and acetone. As shown in Figure 7c, when the two sensors were exposed to methanol vapor at the concentration of 1560 ppm, the *λ_max_* of 3D I-(A-β-CD)-AM PC was blue-shifted by around 43 nm, while the *λ_max_* of the 3D O-(A-β-CD)-AM PCs were blue-shifted by only about 26 nm. The inverse opal structure produced greater volumetric variation. The sensor was placed in HCl solution for elution for 10 min, and then repeatedly washed with ultrapure water several times. The water on the sensor surface was absorbed and recorded at this time. The *λ_max_* was detected with the same concentration of gas, and the above experimental steps were repeated. As shown in Figure 7d, when methanol is added, the reflection peak is 588 nm with a coefficient of variation of 0.2%, and when methanol is discharged, the reflection peak is 548 nm with a coefficient of variation of 0.1%. After six cycles of testing, the 3D I-(A-β-CD)-AM PCs maintained a constant response to methanol.

### 2.5. 3D (A-β-CD)-AM Detection Mechanism

When conventional photonic crystal hydrogel sensors detect gases, the small amount of gas adsorbed leads to insignificant structural color changes. β-CD is a supramolecular host. Driven by intermolecular forces, β-CD selectively binds to methanol, ethanol, propanol, toluene, xylene, and acetone. The introduction of β-CD modified by acryloyl chloride to photonic crystal hydrogels enhances the adsorption of gases and achieves significant structural color changes. Β-CD is similar with amplifiers and increases the displacement of *λ_max_*. 3D O-(A-β-CD)-AM PCs and 3D I-(A-β-CD)-AM PCs detect gases on slightly different principles. 3D O-(A-β-CD)-AM PCs are used to detect toluene, xylene, and acetone. These gases are adsorbed on the surface of colloidal microspheres. As the gas concentration increases, the concentration of gas adsorbed on the surface of the colloidal microsphere increases and the ERI increases. While the internal cavity of β-CD shows hydrophobicity, the greater the hydrophobicity, the more gases enter the system and can be adsorbed on the surface of colloidal microspheres. The hydrophobicity is xylene, toluene, and acetone in descending order. Therefore, the maximum redshift of *λ_max_* is xylene, toluene, and acetone in descending order. 3D I-(A-β-CD)-AM PCs are used to detect methanol, ethanol, and propanol. Its colloidal microspheres have been removed, and toluene, xylene, and acetone cannot be adsorbed on the surface of the colloidal microspheres. It follows that the redshift of *λ_max_* is very small, but the maximum redshift of *λ_max_* is still xylene, toluene, and acetone in descending order. Methanol, ethanol, and propanol cause the hydrogel volume to shrink, the lattice spacing to decrease, and *λ_max_* is blue-shifted. Their hydrophobicity is: methanol < ethanol < propanol. However, before entering the hydrogel system, the gases go through the hydrogel water environment. Compared to ethanol and propanol, methanol is easier to enter the system. It is obvious that the largest blue shift of the *λ_max_* is observed in methanol gas.

### 2.6. Response of 3D O-(A-β-CD)-AM PC to Stretching Ability

In addition to high selectivity, the sensor also requires a certain tensile property [47,48,49]. The PCs had an outstanding stretching ability and the structural color changed with the different pulling forces. As the strain increased from 0% to 64%, the structural color of 3D O-(A-β-CD)-AM PCs with a diameter of 225 nm PMMA gradually changed from red to blue-green (Figure 8a). As shown in Figure 8b, the *λ_max_* was blue-shifted from 659 nm to 553 nm as the pulling force increased. The increase in the spacing of the colloidal microspheres in the horizontal direction was smaller than the decrease in the vertical direction, so the reflection peak was blue-shifted. In addition, as the pulling force increased, the amount of change in the reflection peak decreased gradually (Figure 8c). With the application of the pulling force, the reduction of the distance between the colloidal microspheres in the vertical direction became smaller, so a larger force was applied to achieve the same blue shift in the large pulling force range. To test the practicality of the material, a stretching cycle experiment was carried out. The 3D O-(A-β-CD)-AM PC formed by in-situ polymerization and cross-linking reaction with acrylamide could effectively dissipate energy during ten cycles and maintained the integrity of the gel polymer segment. In the process of cyclic stretching, the 3D O-(A-β-CD)-AM PCs exhibited a structural color change of red-blue green-red, and the corresponding reflectance spectrum is shown in Figure 8d. With the change of strain, the 3D O-(A-β-CD)-AM PC can respond quickly, and the reflection peak changed from 659 nm to 553 nm. Due to the good mechanical properties of the 3D O-(A-β-CD)-AM PC, the reflection peak can be instantly restored to 659 nm after the pulling force is removed.

## 3. Conclusions

In summary, a PC sensor is formed by free radical polymerization of acryloyl chloride-modified β-CD and 3D PMMA colloidal microsphere arrays. Using the adsorption ability of β-CD to gas, 3D (A-β-CD)-AM PC is used to detect VOC gases. 3D I-(A-β-CD)-AM PC was used to detect toluene, xylene, and acetone, which had the most obvious response to xylene gas with a red shift of 27 nm at the concentration of 820 ppm. 3D O-(A-β-CD)-AM PC was used to detect methanol, ethanol, and acetone, and was exposed to methanol vapor at a concentration of 1560 ppm and was blue-shifted by around 43 nm. The sensor has broad prospects in the detection of alcohols and the release of alcohol-dissolved drugs.

## 4. Materials and Methods

### 4.1. Chemical Reagents

β-cyclodextrin (β-CD, AR), methyl methacrylate (MMA, AR), acrylamide (AM, AR), N,N’-methylenebisacrylamide (BIS, AR),2,2-diethoxyacetophenone (DEAP, AR), potassium hydroxide (KOH, AR), toluene, xylene, acetone, methanol, ethanol, and propanol were furnished by J&K Scientific. Potassium persulfate (KPS, AR) and sodium dodecyl sulfate (SDS, AR) were purchased from Sinopharm Chemical Reagents Co., Ltd. The MMA was purified by basic alumina. The BIS was purified by recrystallization. Other chemicals were not purified.

### 4.2. Synthesis of A-β-CD

KOH (10 g) and ultrapure water (125 mL) were mixed and stirred at 25 °C. β-CD was added to the solution at 0 °C. Acryloyl chloride was slowly added to the solution in 30 min with the reaction temperature at 0 °C. The mixture was stirred and heated to 40 °C for 6 h. After the reaction, the filtrate was extracted by using a rotary evaporator and the residue was recrystallized three times in acetone.

### 4.3. Preparation of the PMMA Colloidal Particles and Arrays

Poly(methyl methacrylate) (PMMA) colloidal microspheres were synthesized by an emulsion-free polymerization method. The quantitative MMA and ultrapure water (280 mL) were added to a 500 mL four-necked flask, respectively. The mixture was stirred at 300 rpm and heated to 80 °C under nitrogen. KPS immersed in ultrapure water (10 mL) was added to the four-necked flask. The solution was heated and stirred for 45 min at 80 °C. The resulting product was centrifuged at 9000 rpm and washed with ultrapure water three times. The formulation is shown in Table 1.

3D PMMA photonic crystal arrays were prepared by vertical deposition. The as-prepared PMMA colloidal microsphere solution was diluted to 0.2 wt% and glasses were cleaned with water, ethanol, and plasma cleaner. The glasses were placed vertically in the container and immersed in the solution. The container was left to stand at 30 °C and 50% relative humidity. After two or three days, 3D PMMA arrays were acquired.

### 4.4. Fabrication of 3D O-(A-β-CD)-AM PhCs and 3D I-(A-β-CD)-AM PhCs

First, hydrogel precursors were prepared. A-β-CD was added to 8 mL ultrapure water with vigorous stirring at 25 °C. After the solution was clear, AM, BIS, and DEAP were added in sequence. Hydrogel formulations are shown in Appendix A. The mixture was stirred for 10 min and left to remove air bubbles. The chemical structure of the hydrogel is formed by free radical polymerization. Then, 3D I-(A-β-CD)-AM PC was fabricated. About 100 μL hydrogel precursors were added around the array and immersed in the array by capillary action. The “sandwich structure” was placed in a UV crosslinker with a wavelength of 365 nm and photopolymerized for 8 min. The 3D O-(A-β-CD)-AM PC was removed from the glasses, washed with ultrapure water three times, and finally cut into rectangles of similar sizes. Last, 3D I-(A-β-CD)-AM PC was prepared. The 3D O-(A-β-CD)-AM PC was soaked in toluene solution until the color no longer changed to fully etch the PMMA templates and 3D I-(A-β-CD)-AM PC was obtained. The cleaning process was the same as 3D O-(A-β-CD)-AM PC.

### 4.5. Materials Characterization

The detailed microstructure was recorded on field-emission scanning electron microscopy (SEM; Hitachi JSM-7500F; JEOL; Tokyo, Japan). Fourier transform infrared (FTIR) spectra were carried out by a Thermo Fisher Scientific Nicolet IS 10 FT-IR spectrometer (Thermo Scientific; Waltham, MA, USA) with an average of 64 scans. The hydrogen spectrum and carbon spectrum of A-β-CD were detected by Avance III HD 400 MHz (Bruker; Billerica, MA, USA). Optical photos of the structural color were taken with a Nikon D3500 digital camera. The optical properties of materials were measured by Avantes A-2048TEC (Avantes; Beijing, China).

## Figures and Tables

**Figure 1 gels-09-00083-f001:**
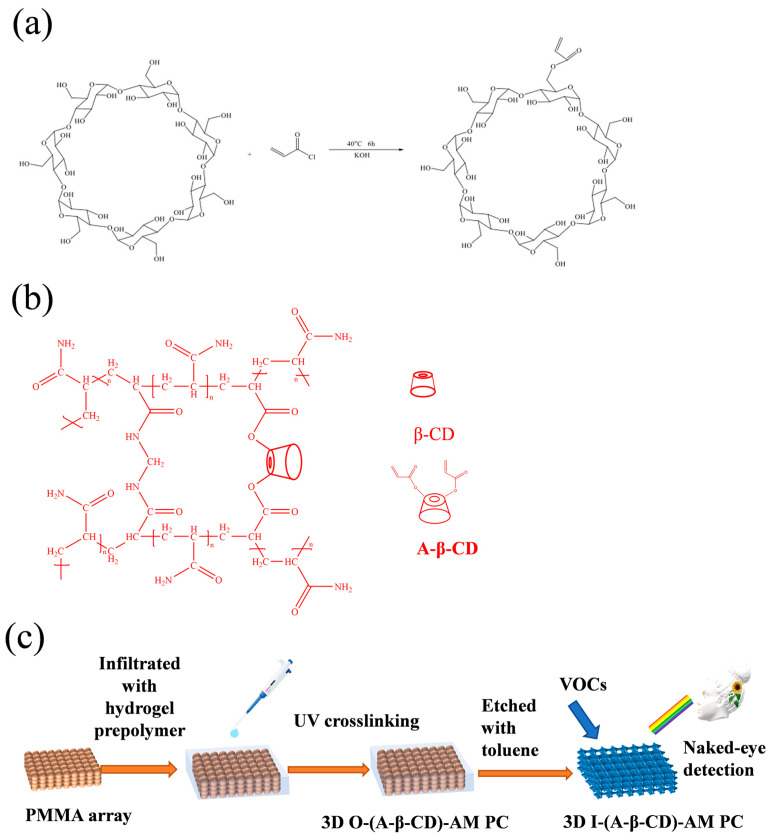
(**a**) The synthetic route of A-β-CD; (**b**) the chemical structure of the hydrogel formed by free radical polymerization; (**c**) schematic illustration of the (A-β-CD)-AM PC preparation.

**Figure 2 gels-09-00083-f002:**
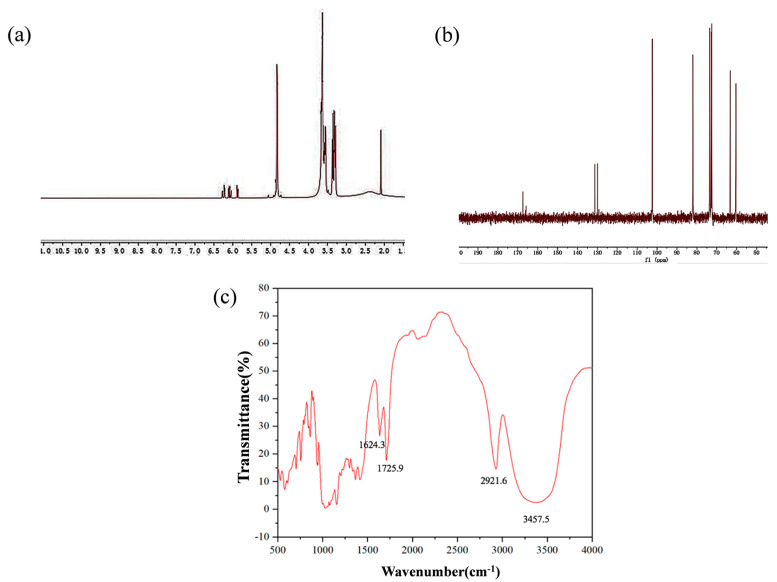
(**a**) ^1^H NMR; (**b**) ^13^C NMR; (**c**) FTIR of A−β−CD.

**Figure 3 gels-09-00083-f003:**
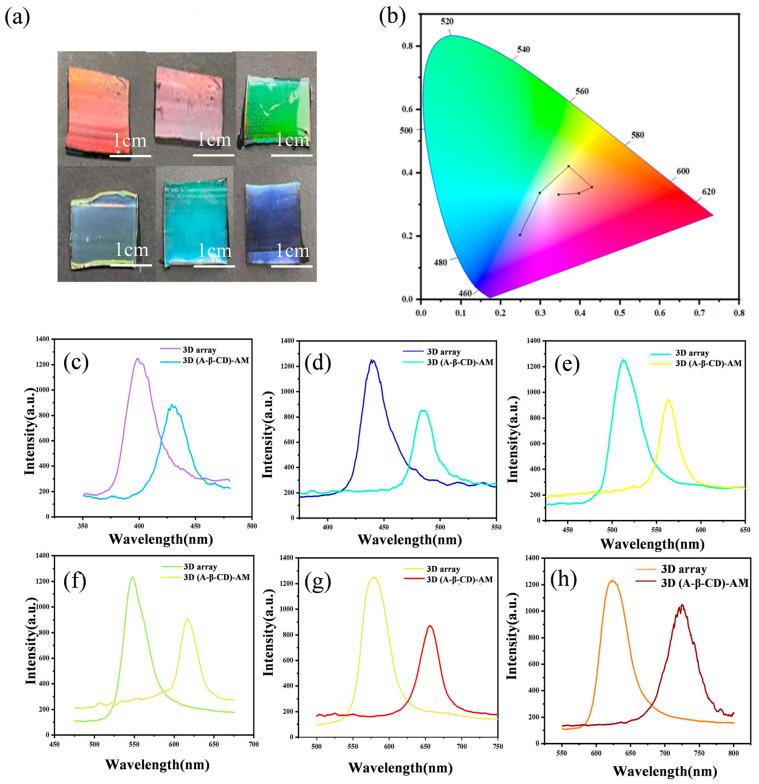
(**a**) Optical photos of 3D O-(A-β-CD)-AM PCs of PMMA with different diameters; (**b**) CIE of 3D O-(A-β-CD)-AM PCs of PMMA with different diameters; (**c**–**h**) the reflection peak of 3D O-(A-β-CD)-AM PCs and PMMA arrays.

**Figure 4 gels-09-00083-f004:**
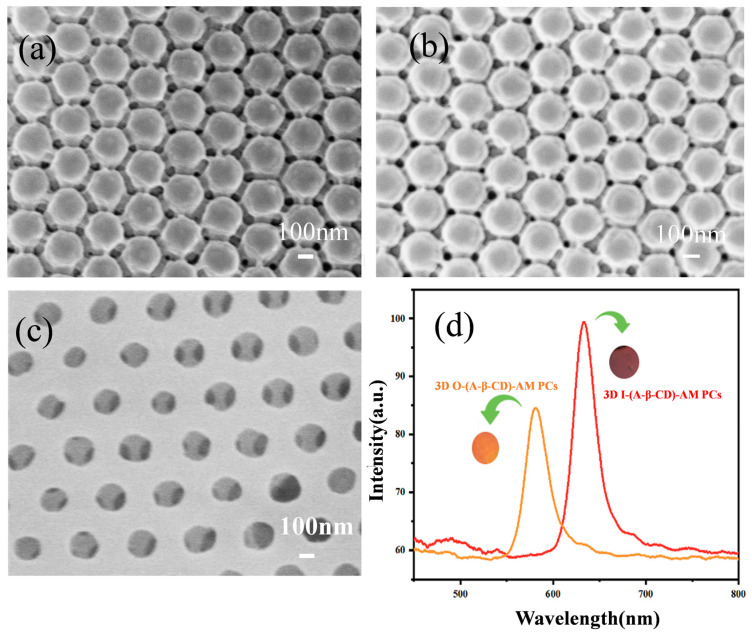
SEM images of 3D O-(A-β-CD)-AM PCs with PMMA of (**a**) 225 nm; (**b**) 240 nm; (**c**) 3D O-(A-β-CD)-AM PCs; (**d**) the reflection peak of 3D O-(A-β-CD)-AM PC and 3D I-(A-β-CD)-AM PC.

**Figure 5 gels-09-00083-f005:**
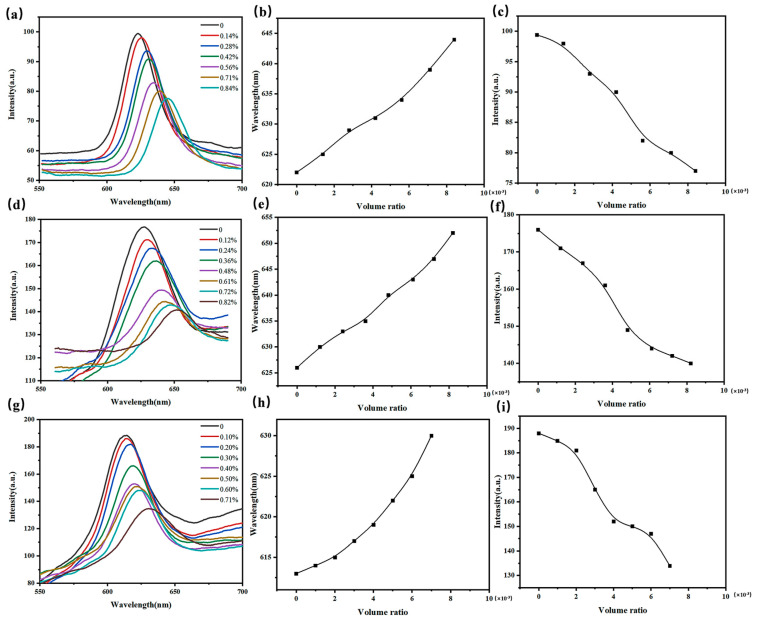
Responses of the 3D O-(A-β-CD)-AM PC to VOC gas. (**a**) *λ_max_* responses of 3D O-(A-β-CD)-AM PC to toluene gas; (**b**) the relationship between the wavelength of *λ_max_* and volume ratio; (**c**) the relationship between the intensity of *λ_max_* and volume ratio of the toluene gas; (**d**–**f**) *λ_max_* responses of 3D O-(A-β-CD)-AM PC to xylene gas; (**g**–**i**) *λ_max_* responses of 3D O-(A-β-CD)-AM PC to acetone gas.

**Figure 6 gels-09-00083-f006:**
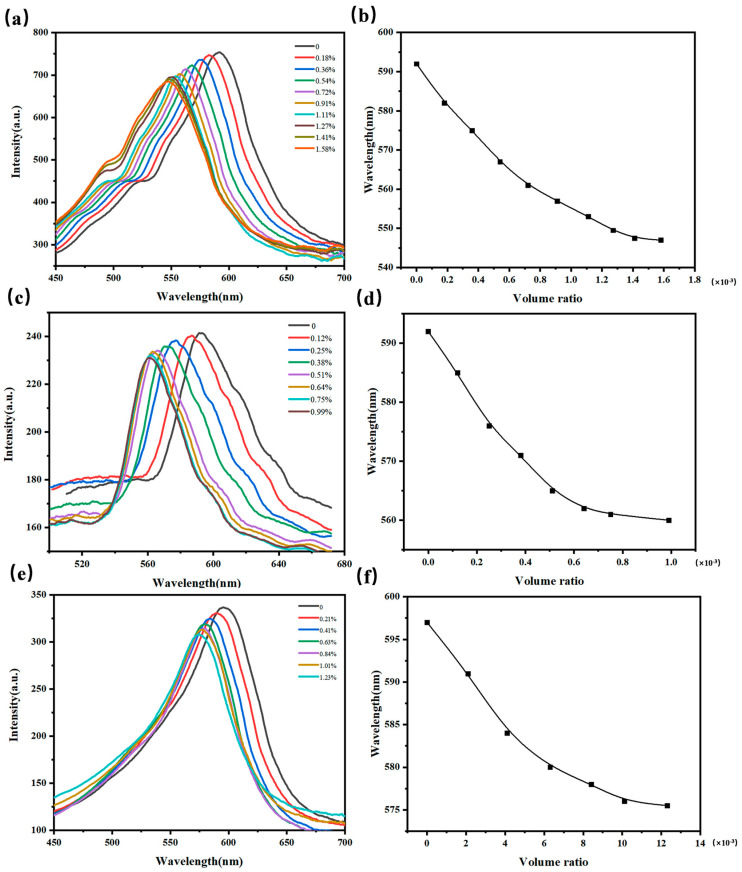
Responses of the 3D I-(A-β-CD)-AM PCs to VOC gas. (**a**) *λ_max_* responses of 3D I-(A-β-CD)-AM PCs to methanol gas; (**b**) the relationship between the wavelength of *λ_max_* and volume ratio; (**c**,**d**) *λ_max_* responses of 3D I-(A-β-CD)-AM PCs to ethanol gas; (**e**,**f**) *λ_max_* responses of 3D I-(A-β-CD)-AM PCs to propanol gas.

**Figure 7 gels-09-00083-f007:**
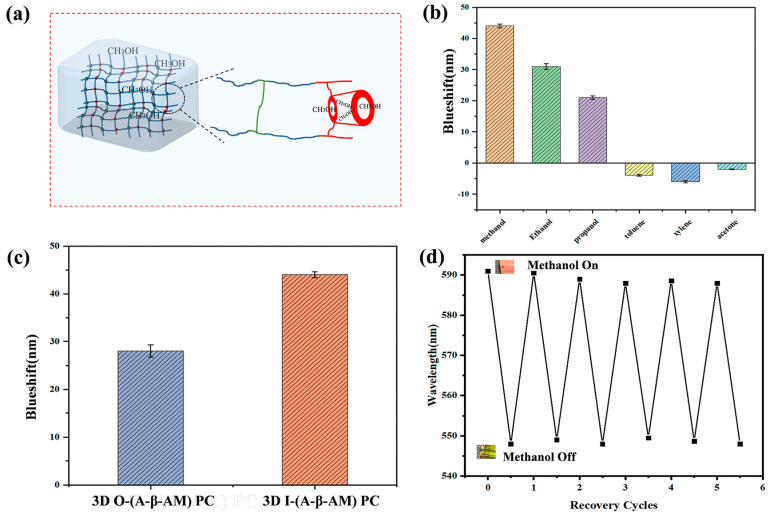
(**a**) Schematic diagram of methanol entering the interior of the sensor; (**b**) the responses of vapors by 3D O-(A-β-CD)-AM PC; (**c**) comparison of 3D O-(A-β-CD)-AM PC and 3D I-(A-β-CD)-AM PC to methanol vapor; (**d**) cyclability of 3D I-(A-β-CD)-AM PC.

**Figure 8 gels-09-00083-f008:**
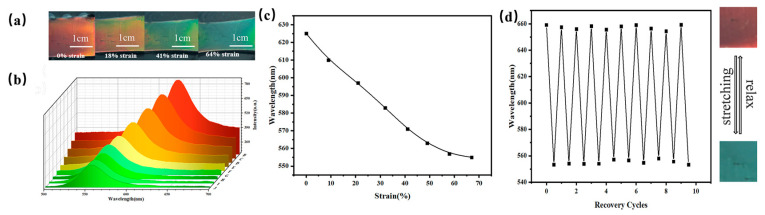
(**a**) Optical photograph of 3D O-(A-β-CD)-AM PC during stretching; (**b**) corresponding reflection spectra during stretching; (**c**) variation of reflected wavelength with increasing tensile force; (**d**) cyclability of 3D O-(A-β-CD)-AM PC.

**Table 1 gels-09-00083-t001:** PMMA formulation.

Number	MMA (mL)	KPS (g)	Diameter (nm)
1	9	0.9	165
2	10	0.6	180
3	11	0.5	210
4	12	0.4	225
5	18	0.4	240
6	20	0.4	255

## Data Availability

The data presented in this study are available on request from the corresponding author.

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
