# Peer review of "Specific Alcohol-Responsive Photonic Crystal Sensors Based on Host-Guest Recognition"

_gels, 2023, doi:10.3390/gels9020083_

Round 1
Reviewer 1 Report
Manuscript ID: gels-2152412
Article Title: Specific alcohol-responsive photonic crystal sensors based on Host-Guest Recognition Authors: Xiaolu Cai, Xiaojing Zhang, Jing Fan, Wenxiang Zheng, Tianyi Zhang, Lili Qiu *, Zihui Meng *
Comments and Suggestions for Authors
This manuscript presents the photonic crystal sensors based on the three dimension of poly (methyl methacrylate) (PMMA) colloidal microsphere are presented. The photonic sensing based on the change of balanced reflection of photonic structures can hardly distinguish chemical species with similar refractive indices. The 3D O-(A-β-CD)-AM PC was applied to VOCs gas sensing such as toluene, xylene, acetone, methanol, ethanol and propanol. The detection of VOCs gas can be able to distinguish alcohol species by the visible spectrum shift. However, the sensors should provide the specification such the sensitivity, limit of detection and limit of quantification. The image quality is poor and unclear. If possible, increase the resolution of the images and graphs. The RBC color characteristic could represent the color changing by absorbed VOCs in a surface of sensor, the detection of color can be detected by image processing. In this work have a significant contribution to the field and is suitable for publication.
(1) Figure 1 (a-c), line 68. The picture quality should be increased the resolution.
(2) In 2. Results and Discussion, line 85-87. The author said that the NMR and FTIR spectrum was compared with the literature, please add the cited number.
(3) Figure 2 (a-c), line 88. The picture quality should be increased the resolution.
(4) Figure 3 (a-h), line 107. The picture quality should be increased the resolution.
(5) In line 127-129. The value of material refractive index was tested, or it was obtained from references from other sources, please detail.
(6) Figure 4 (a-d), line 132. The picture quality should be increased the resolution.
(7) In line 135-141, the information regarding volatile organic compounds should be in the introduction more than the results and discussion.
(8) In line 148-149, the author said that “As a comparison, when Wang from meng group detected SO2 through opal photonic 148 crystal cellulose membrane, the red shift of the reflection peak was only 7 nm.”, please add the cited number and detail what’s the difference between author’s work and Wang group.
(9) In a last paragraph of 2.3, the response of 3D O-(A-β-CD)-AM PCs to VOCs should be discussed in term of the concentration (ppm) as a function both wavelength shift and intensity. Moreover, the limit of detection and limit of quantitation should be explained.
(10) Figure 5 (a-i), line 154. The picture quality should be increased the resolution.
(11) In 2.4, the Response of 3D I-(A-β-CD)-AM to VOCs should be discussed in term of the concentration (ppm) as a function both wavelength shift and intensity. Moreover, the limit of detection and limit of quantitation should be explained.
(12) Figure 6 (a-f), line 167. The picture quality should be increased the resolution.
(13) In a last paragraph of 2.4, the reproductivity should be discussed until the sensor cannot detect the VOCs or the signal decreased until less than 50% of an originated detection.
(14) In line 195, the cycle of testing in the sensor ability, the coefficient of variation (CV) should be used to describe the repeatability of signal.
(15) In conclusions, the photonic crystal sensors should be suggested, if the VOCs is a compound (mixed gas), how we can apply the sensor to detect them.
(16) In materials and methods, 4.3, line 253, the quantitative MMA concentration should be detailed.
(17) In materials and methods, 4.3, line 255, KPS immersed in ultrapure water (10 mL) 255 was added to the four-necked flask, the KPS concentration should be detailed.
(18) In materials and methods, 4.3, line 255, the initiator concentration for PMMA sub-micron particle synthesis should be detailed.
(19) In materials and methods, 4.3, line 263, the author said that “After two or three days, 3D PMMA arrays were acquired.” After two or three days, how can know the day that suitable? The manuscript should detail, such as the humidity or which scientific parameter.
(20) In materials and methods, 4.4, line 265, the author said that “A-β-CD was added into 8 mL ultrapure 265 water with vigorous stirring at 25 C”. A-β-CD concentration should be detailed.
(21) In materials and methods, 4.4, line 271, the author said that “The “sandwich structure” was placed in a UV crosslinker with a wavelength of 365 nm and photopolymerized for 8 minutes.”. The energy dose of photopolymerizedwith 8 minutes, how much of energy in the unit of mJ?
(22) In materials and methods, 4.4, line 272-273, the author said that “The 3D O-(A-β-CD)-AM PC was removed from the glasses, washed with ul trapure water three times, and finally cut into rectangles of the same size.”. the same size, how much of size?
Reviewer 2 Report
Cai et al reported a PC sensor which was synthesized by free radical polymerization of acryloyl chloride-modified β-CD and 3D PMMA colloidal microsphere arrays. 3D I-(A-β-CD)- AM PC was used to detect toluene, xylene and acetone, which had the most obvious response to xylene gas with a red shift of 27 nm at the concentration of 820 ppm. 3D O-(A-β-CD)-AM PC was used to detect methanol, ethanol and acetone, which exposed to methanol vapor at the concentration of 1560 ppm was blue-shifted by around 43 nm.The topic is interesting and fit the journal scope well, however, some defects need be cleared before processing the manuscript further:
1. The authors do not clearly discuss what we have learned from this work about general material design and how we can use their results to make structures with desired prescribed properties. Comments about these issues appear here and there in the paper, but I believe there should be a designated paragraph or to address this extensively. Some related questions are, why did the authors choose the specific materials? what is the mechanism of VOC absorption? The chemical – physical assessments on the materials structure seems incomplete.
2. The figure contents need be polished, there are lots of issues as,
2.1 In most Figures, the font seems too small to see, please check and revise;
2.2 In Fig3a and Fig 8a, please provide the scale bar;
2.3 In Fig.4, please re-do the SEM, as the resolution for current images are too low.
3. In 2.5, the demonstration of Response of 3D O-(A-β-CD)-AM PC to Stretching Ability, needs more quantitated details, the responding time has not been well examined. The mechanical input based color change generation seems very slim.
4. Some references might be good to strengthen the following sections:
4.1 Responsive color sensor
a. https://pubs.acs.org/doi/abs/10.1021/acsami.2c16938
b. https://www.nature.com/articles/s41467-020-15288-8
c. https://link.springer.com/article/10.1007/s42114-022-00447-w
4.2 Stretchability testing of sensors
d. https://pubs.acs.org/doi/abs/10.1021/acs.chemmater.1c01246
e. https://link.springer.com/article/10.1007/s42114-021-00292-3
f. https://link.springer.com/article/10.1007/s42114-022-00531-1
5. Another major flaw is that the manuscript is full of typos, mistakes in both grammar and syntax, and in many cases incomprehensible. I will not enumerate all the issues here, but I advise the authors to carefully edit their paper before resubmitting to avoid sloppiness.
In conclusion, I would recommend a major revision.
Round 2
Reviewer 1 Report
The author has explained all points completely. In this work have a significant contribution to the field and is suitable for publication.
Reviewer 2 Report
The revision has addressed my previous comments adequately, I would recommend to accept for publishing.